# Factors indicating intention to vaccinate with a COVID-19 vaccine among older U.S. adults

**Janeta Nikolovski**[1], **Martin Koldijk**[1], **Gerrit Jan Weverling**[1], **John Spertus**[2], **Mintu Turakhia**[3], **Leslie Saxon**[4], **Mike Gibson**[5], **John Whang**[1], **Troy Sarich**[1], **Robert Zambon**[1], **Nnamdi Ezeanochie**[6], **Jennifer Turgiss**[6], **Robyn Jones**[6], **Jeff Stoddard**[1], **Paul Burton**[1], **Ann Marie Navar**[7] *

1 Janssen Pharmaceutical Companies of Johnson & Johnson, Titusville, NJ, United States of America, 2 Department of Internal Medicine, University of Missouri-Kansas City, Kansas City, MO, United States of America, 3 Department of Medicine, Stanford University, Palo Alto, CA, United States of America, 4 Department of Medicine, University of Southern California, Los Angeles, CA, United States of America, 5 Department of Medicine, Harvard, Boston, MA, United States of America, 6 Johnson & Johnson, New Brunswick, NJ, United States of America, 7 Department of Medicine, UT Southwestern Medical Center, Dallas, TX, United States of America

* ann.navar@UTsouthwestern.edu

**Data Availability Statement:** The data gathered is part of the Heartline Study, a collaborative study between Johnson & Johnson and Apple. Due to contractual terms between Johnson & Johnson

## Abstract

### Background

The success of vaccination efforts to curb the COVID-19 pandemic will require broad public uptake of immunization and highlights the importance of understanding factors associated with willingness to receive a vaccine.

### Methods

U.S. adults aged 65 and older enrolled in the Heartline™ clinical study were invited to complete a COVID-19 vaccine assessment through the Heartline™ mobile application between November 6–20, 2020. Factors associated with willingness to receive a COVID-19 vaccine were evaluated using an ordered logistic regression as well as a Random Forest classification algorithm.

### Results

Among 9,106 study participants, 81.3% (n = 7402) responded and had available demographic data. The majority (91.3%) reported a willingness to be vaccinated. Factors most strongly associated with vaccine willingness were beliefs about the safety and efficacy of COVID-19 vaccines and vaccines in general. Women and Black or African American respondents reported lower willingness to vaccinate. Among those less willing to get vaccinated, 66.2% said that they would talk with their health provider before making a decision. During the study, positive results from the first COVID-19 vaccine outcome study were released; vaccine willingness increased after this report.

and Apple, the authors do not have rights to share data. Because the study was conducted by J&J in collaboration with Apple, the authors do have differential access to the data contractually compared with external parties. Data requests can be sent to Simrati Kaul (skaul@its.jnj.com) and will be reviewed by Johnson & Johnson and Apple.

**Funding:** The Heartline Study is supported by Johnson & Johnson (www.jnj.com) and Apple (www.apple.com). Johnson & Johnson was responsible for study design, data collection and analysis, decision to publish and preparation of the manuscript. Apple neither supported nor participated in survey development or interpretation for this article. JS, MT, LS, MG, AMN receive personal fees and/or grants from Janssen.

**Competing interests:** I have read the journal's policy and the authors of this manuscript have the following competing interests: JN, MK, GJW, JW, TS, JS, PB, RZ, NE, JT, RJ are employees of Janssen and Johnson and Johnson. JN, MK, GJW, JW, TS, JS, PB, RZ, NE, JT, RJ are employees of Janssen and Johnson and Johnson. This does not alter our adherence to PLOS ONE policies on sharing data and materials. JS reports personal fees from Amgen, personal fees from Bayer, personal fees from Merck, personal fees from Novartis, personal fees from Janssen, personal fees from Myokardia, personal fees from Blue Cross Blue Shield of Kansas City, outside the submitted work. In addition, Dr. Spertus has a patent Copyright to the KCCQ with royalties paid and Equity in Health Outcomes Sciences. This does not alter my adherence to PLOS ONE policies on sharing data and materials. MT reports grants from Janssen Inc, personal fees from Medtronic Inc, personal fees from Abbott, grants from Boehringer Ingelheim, grants and personal fees from Cardiva Medical, personal fees from iRhythm, grants from Bristol Myers Squibb, grants from American Heart Association, grants from SentreHeart, personal fees from Novartis, personal fees from Biotronik, personal fees from Sanofi, personal fees from Pfizer, grants from Apple, grants from Bayer, personal fees from Myokardia, personal fees from Johnson & Johnson, personal fees from Milestone Pharmaceuticals, outside the submitted work; and Dr. Turakhia is an editor for JAMA Cardiology. This does not alter my adherence to PLOS ONE policies on sharing data and materials. LS reports personal fees from J&J, outside the submitted work. This does not alter my adherence to PLOS ONE policies on sharing data and materials. MG receives research grant support from Janssen. This does not alter my adherence to PLOS ONE policies on sharing data and materials. AMN receives

## Conclusions

Even among older adults at high-risk for COVID-19 complications who are participating in a longitudinal clinical study, 1 in 11 reported lack of willingness to receive COVID-19 vaccine in November 2020. Variability in vaccine willingness by gender, race, education, and income suggests the potential for uneven vaccine uptake. Education by health providers directed toward assuaging concerns about vaccine safety and efficacy can help improve vaccine acceptance among those less willing.

## Trial registration

Clinicaltrials.gov NCT04276441.

## Introduction

With the recent FDA Emergency Use Authorization (EUA) for three COVID-19 vaccines, significant attention is now being placed on whether sufficient numbers of the public will be willing to be immunized to control the pandemic and how to ensure the public is adequately informed about the vaccine [1–3]. Surveys in the United States showed an initial decline in reported willingness to receive a COVID-19 vaccine [4–9], though some recent data appear more promising [5, 10, 11].

Individual perceptions about vaccines and about COVID-19 can strongly influence the decision to vaccinate against COVID-19 and are likely more associated with vaccine behaviors than demographics alone [9, 12–14]. Vaccine uptake in older adults is of particular importance as increasing age is the leading risk factor for mortality and complications from COVID-19 infections [15, 16]. Guidelines have been developed that identify priority populations for vaccination (CDC.gov), with recommendations including those older than 65 years. Understanding who is least likely to vaccinate as well as potentially modifiable factors influencing their decisions, is important to develop public health strategies to overcome vaccine hesitancy, especially among older and higher risk populations [15].

Smartphone-based research can facilitate rapid data collection for timely research questions. To accelerate public health's understanding of current perspectives on vaccinations, we surveyed subjects already participating in a smartphone-based clinical trial, the Heartline™ Study, a virtual clinical study that is enrolling U.S. adults age 65 years and older. These participants are at higher risk of COVID-19 related morbidity and mortality due to their age, and thus have a large potential benefit from immunization. We hypothesized that older adults in our sample population would have a high willingness to vaccinate and sought to understand factors associated with those less willing. In order to understand factors associated with and indicative of willingness to vaccinate in this higher risk population, we deployed a vaccine survey to all participants through the Heartline™ platform in November, 2020.

## Materials and methods

### Study population and data collection

On February 25, 2020, enrollment began into The Heartline™ Study (clinicaltrials.gov, NCT04276441 and https://heartline.com/), a large heart health clinical study in the United States. Eligibility requirements include age 65 and older, possessing an iPhone 6s or later, current Original Medicare beneficiary, U.S. resident, and English-speaking (see clinicaltrials.gov,

consulting fees from Janssen for serving on the steering committee of the Heartline study. In addition, Dr. Navar receives funding for research to her institution from Amgen and Janssen, and honoraria and consulting fees from Amarin, Amgen, Astra Zeneca, BI, Esperion, Lilly, Sanofi, Regeneron, NovoNordisk, Novartis, The Medicines Company, New Amsterdam, Cerner, 89Bio, and Pfizer, outside the scope of this work. This does not alter my adherence to PLOS ONE policies on sharing data and materials.

NCT04276441 for complete inclusion and exclusion criteria). The Heartline[TM] Study is investigating if wearable and custom-built mobile app technologies can enable earlier detection of atrial fibrillation (AF), reduce the incidence of clinical events, and improve adherence with oral anticoagulants in those with AF. The study is completely virtual, without the need for study site visits, and is conducted on the Heartline[TM] app. The study platform is designed to introduce novel surveys as needed throughout the study. The study protocol was approved by Western Institutional Review Board and all participants submit a signed informed consent to analyze de-identified data through the Heartline[TM] app.

Willingness to vaccinate against COVID-19 was assessed through an optional survey through the Heartline[TM] app between November 6 and 20, 2020. The assessment was framed by the World Health Organization recommended Capability, Opportunity, Motivation and Behavior model (COM-B) model for addressing vaccine hesitancy and acceptance [17]. The assessment included questions regarding beliefs about vaccines in general, beliefs about COVID-19 and the COVID-19 vaccine, and opinions on vaccine dosing and potential side effects (see S1 Table for the complete assessment). The assessment was offered to all study participants. Demographic data were collected at the time of study enrollment, which included race and gender by self-report. The primary outcome of the study was self-reported willingness to receive a COVID-19 vaccine. This was captured on a four-point scale (very willing, somewhat willing, not very willing, not at all willing).

## Statistical analysis and model

To evaluate factors associated with vaccine willingness, we applied two different analytic approaches. First, determinants of willingness to be vaccinated were evaluated using an ordered univariate logistic regression model with the 4 levels of willingness to be vaccinated as the outcome while adjusting for race (Asian, Black or African American, White and other) and gender. Next, to identify a set of determinants to separate those who are willing to vaccinate from those who are not, recursive feature elimination in combination with a Random Forest classification algorithm [18] was performed. In this analysis, willingness to vaccinate was classified into two categories: willing to vaccinate (combining very willing with somewhat willing) and not willing to vaccinate (combining not at all willing with not very willing). The algorithm was trained using a dataset that was randomly split by stratum (either not willing or willing to be vaccinated), into a 2/3 training set (n = 4935 with 91.3% willing to vaccinate) and a 1/3 different hold-out set to verify model performance (n = 2467 with 91.3% willing to vaccinate).

For the machine learning models, categorical question and answer combinations from the survey and demographic variables, including gender, age, body mass index (BMI), race, income and education, were One-Hot encoded (i.e. converted to dummy variables). This resulted in a total of 85 features (S2 Table) that were used for the construction of the Random Forest. The model was tuned with respect to terminal node size and mtry (number of features available for splitting at each tree node), with class imbalance being addressed using stratified re-sampling for each of the 4001 trees. The most relevant features were determined using a recursive feature elimination approach. Starting with a model trained using the complete set of features, normalized permutation importance scores were determined and the bottom 4% of features were removed. The resulting model with all features (n = 85) or the recursively reduced models were evaluated using the hold-out dataset. The model with the minimum set of features was selected based on maintaining a high balanced accuracy on the hold-out dataset (i.e., the average of specificity and sensitivity), compared to the full model.

Shortly after the November survey was offered, Pfizer announced a first interim analysis reporting >90% efficacy of their vaccine candidate on November 9, 2020 [19]. We investigated

if that news had an impact on willingness to vaccinate in our population and compared those who answered the survey before the 9th to those who answered after (excluding those who answered on the 9th itself, and those who answered on or after November 16th, the Moderna announcement [20]) using an ordered logistic regression model with four-level willingness to be vaccinated as the outcome and an indicator variable for the time period (before and after November 9th) and gender as predictors. Statistical analyses were performed using *R* (3.5.1 and 3.6.1) and r-packages *randomForest* (4.6–14) [18], *caret* (6.0–84) [21] and *MASS* (7.3–53) [22].

## Results

### Willingness to vaccinate against COVID-19

The assessment was offered to 9,106 participants with 7,621 (83.6%) completing the survey. Excluding participants (n = 219) missing demographic data left 7,402 participants (81.3%) for the study analyses (Table 1). Overall, 63.6% of participants reported they were very willing to receive a COVID-19 vaccine, 27.8% were somewhat willing, 6.0% were not very willing, and 2.6% were not at all willing. Treated as a dichotomous response, 91.3% were considered "willing" to be immunized while 8.7% were "unwilling" (Table 1).

Fig 1 shows results of univariable evaluations of demographic factors associated with vaccine willingness. Black or African American race was most strongly associated with decreased odds of vaccine willingness (odds ratio 0.24, 95% CI 0.18–0.31) (Fig 1). A total of 26.8% of Black or African American participants noted they were not very willing or not at all willing to vaccinate, compared with 8.0% of white participants (Table 1). Women were also less willing to be vaccinated as shown in Table 1 by 12.1% of women and 5.7% of men being not very or not at all willing to vaccinate against COVID-19 (odds ratio 0.49, 95% CI 0.45–0.54) (Fig 1). Income and education are also associated with willingness to be vaccinated, with higher income and higher education being associated with a higher willingness to be vaccinated (Fig 1).

Surveyed beliefs about COVID-19 and the COVID-19 vaccine (Fig 2), as well as beliefs about vaccines in general (S1 Fig), were found to be strongly associated with willingness to vaccinate. Regarding COVID-19, the most strongly associated beliefs included that the COVID-19 vaccine will help protect "myself and others" (odds ratio 38.6, 95% CI 32.4–46.1), the COVID-19 vaccine would be safe and effective (odds ratio 21.6, 95% CI 18.9–24.7), and being comfortable with short term side effects such as prolonged injection site pain (odds ratio 10.9, 95% CI 9.1–13.1). These beliefs were consistently important across participants who were White, Asian, African American or Black (S2 Fig).

### Information seeking

The vast majority of those who would be willing to vaccinate indicated they would talk to their healthcare provider (HCP) or staff before deciding whether or not to receive the vaccine (91.4% of women and 88.9% of men) (S3 Table). The majority of those who indicated they would not be willing to be vaccinated (68.4% of women and 62.3% of men) also indicated they would talk to their healthcare provider before deciding (S3 Table).

Specifically, we categorized the proportions of those who would talk to their HCP before deciding among those not at all willing and not very willing, by gender and race (Table 2). Of those not very willing, greater than 75% would talk to their HCP first (75.0% White, N = 376; 85.3% Black or African American, N = 34). Even among those not at all willing to vaccinate, around half would talk to their HCP before deciding (48.2% White, N = 168; 53.3% Black or African American, N = 15).

**Table 1. Demographics of the participants completing the vaccine assessment overall and stratified by vaccine willingness.**

| | | Total | Unwilling to Receive Vaccine | Willing to Receive Vaccine |
|---|---|---|---|---|
| Subjects (N, %) | | 7402 (100%) | 642 (8.7%) | 6760 (91.3%) |
| Gender (N, %) | Women | 3423 (46.2%) | 414 (12.1%) | 3009 (87.9%) |
| | Men | 3979 (53.8%) | 228 (5.7%) | 3751 (94.3%) |
| Age (yr, mean ± SD) | | 70.8 ± 4.7 | 70.2 ± 4.4 | 70.9 ± 4.8 |
| Age (N, %) | [65,70) | 3509 (47.4%) | 329 (9.4%) | 3180 (90.6%) |
| | [70,75) | 2378 (32.1%) | 216 (9.1%) | 2162 (90.9%) |
| | [75,80) | 1093 (14.8%) | 73 (6.7%) | 1020 (93.3%) |
| | [80,85) | 337 (4.6%) | 18 (5.3%) | 319 (94.7%) |
| | [85,90) | 71 (1.0%) | 6 (8.5%) | 65 (91.5%) |
| | [90,95) | 12 (0.2%) | 0 (0.0%) | 12 (100%) |
| | [95,100) | 2 (0.0%) | 0 (0.0%) | 2 (100%) |
| BMI (kg/m$^2$, mean ± SD) † | | 27.5 ± 5.3 | 27.9 ± 5.6 | 27.4 ± 5.3 |
| BMI (N, %) | Under weight (<18.5) | 90 (1.2%) | 4 (4.4%) | 86 (95.6%) |
| | Normal weight (18.5–25) | 2525 (34.1%) | 209 (8.3%) | 2316 (91.7%) |
| | Overweight (25.0–30) | 2850 (38.5%) | 229 (8.0%) | 2621 (92.0%) |
| | Obese (> = 30) | 1937 (26.2%) | 200 (10.3%) | 1737 (89.7%) |
| Race (N, %) ‡ | American Indian or Alaska Native | 21 (0.3%) | 1 (4.8%) | 20 (95.2%) |
| | Asian | 238 (3.2%) | 21 (8.8%) | 217 (91.2%) |
| | Black or African American | 183 (2.5%) | 49 (26.8%) | 134 (73.2%) |
| | Native Hawaiian or Other Pacific Islander | 8 (0.1%) | 3 (37.5%) | 5 (62.5%) |
| | Two or more | 44 (0.6%) | 3 (6.8%) | 41 (93.2%) |
| | White | 6794 (91.8%) | 544 (8.0%) | 6250 (92.0%) |
| | Prefer not to answer | 114 (1.5%) | 21 (18.4%) | 93 (81.6%) |
| Education (N, %) | Some high school or less | 18 (0.2%) | 3 (16.7%) | 15 (83.3%) |
| | High school diploma or equivalent (GED) | 280 (3.8%) | 41 (14.6%) | 239 (85.4%) |
| | Some college education | 1104 (14.9%) | 152 (13.8%) | 952 (86.2%) |
| | Associate degree (e.g. AA, AS) | 509 (6.9%) | 66 (13.0%) | 443 (87.0%) |
| | Bachelor's degree (e.g. BA, BS) | 2285 (30.9%) | 173 (7.6%) | 2112 (92.4%) |
| | Master's degree (e.g. MA, MS, MEd) | 2179 (29.4%) | 148 (6.8%) | 2031 (93.2%) |
| | Doctorate (e.g. PhD, EdD) | 480 (6.5%) | 29 (6.0%) | 451 (94.0%) |
| | Professional degree (e.g. MD, DDS, DVM) | 499 (6.7%) | 23 (4.6%) | 476 (95.4%) |
| | Prefer not to answer | 48 (0.6%) | 7 (14.6%) | 41 (85.4%) |
| Income (N, %) | Under $30,000 | 356 (4.8%) | 56 (15.7%) | 300 (84.3%) |
| | $30,000-$39,999 | 308 (4.2%) | 43 (14.0%) | 265 (86.0%) |
| | $40,000-$49,999 | 343 (4.6%) | 55 (16.0%) | 288 (84.0%) |
| | $50,000-$59,999 | 442 (6.0%) | 45 (10.2%) | 397 (89.8%) |
| | $60,000-$74,999 | 747 (10.1%) | 79 (10.6%) | 668 (89.4%) |
| | $75,000-$99,999 | 1115 (15.1%) | 78 (7.0%) | 1037 (93.0%) |
| | $100,000-$149,999 | 1485 (20.1%) | 85 (5.7%) | 1400 (94.3%) |
| | $150,000-$200,000 | 688 (9.3%) | 30 (4.4%) | 658 (95.6%) |
| | Above $200,000 | 632 (8.5%) | 26 (4.1%) | 606 (95.9%) |
| | Prefer not to answer | 1286 (17.4%) | 145 (11.3%) | 1141 (88.7%) |

Plus-minus values are means ± standard deviations (SD). Overall column percentages represent % of overall sample (column percent). Percentages in willing and unwilling columns represent row %.

† The body-mass index is the weight in kilograms divided by the square of the height in meters.

‡ Race was reported by the participants, who could select more than one category.

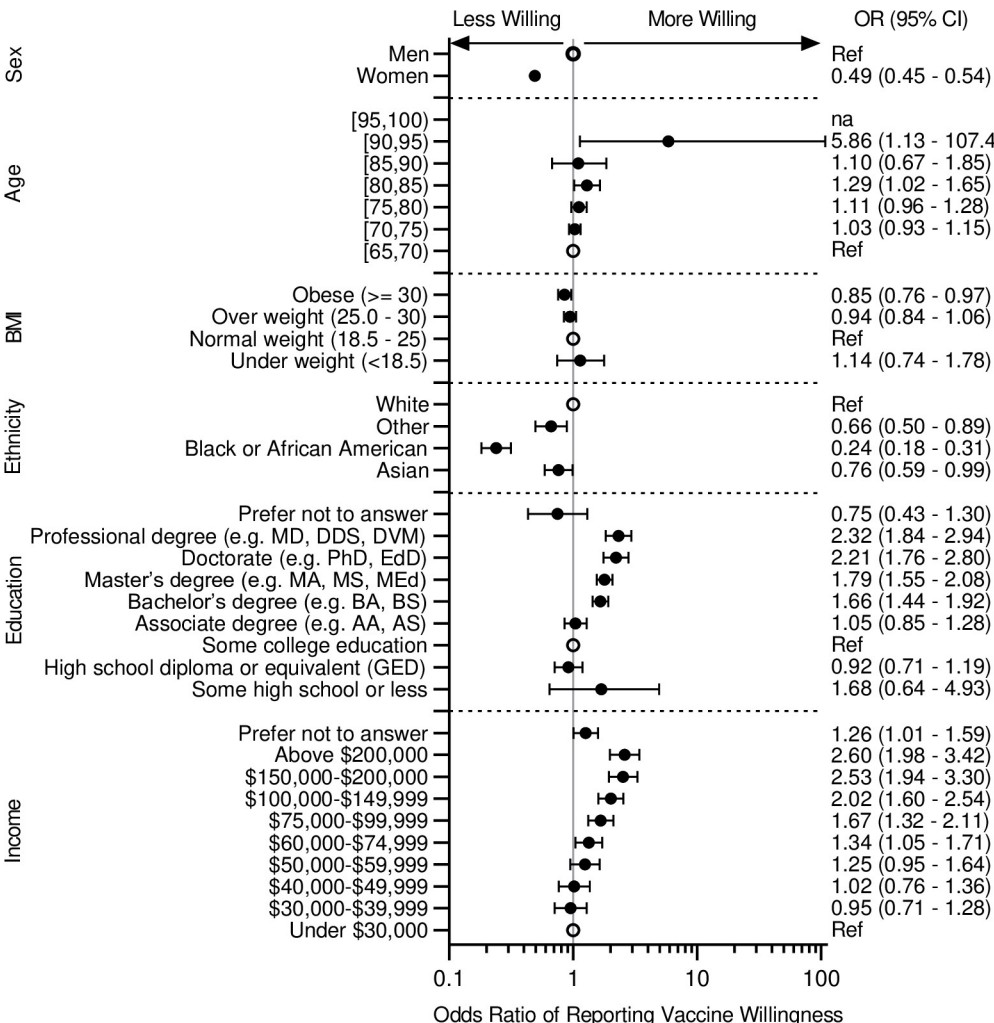

**Fig 1. Forest plot of willingness to vaccinate by demographics.** Shown are Odds Ratios (95% CI) for willingness to vaccinate for the different demographic characteristics. Odds Ratios were calculated using ordered logistic regression model with the 4 levels of willingness to be vaccinated as the outcome while adjusting for gender and race. Reference for each category is indicated by an open circle. *na* indicate not sufficient subjects for this category. 'Native Hawaiian or Other Pacific Islander', 'Two or more', and 'Prefer not to answer' are combined in 'Other'.

## Participant characteristics and beliefs indicative of vaccine willingness

To identify a set of determinants that separate those who are willing and unwilling to be vaccinated, we constructed a Random Forest classification algorithm. From the survey and the available self-reported demographic data, we extracted a list of features (S2 and S3 Tables) to be used in the first model. This initial model using all 85 features resulted in 90.2% balanced accuracy (average of 90.7% sensitivity and 89.7% specificity) when applied to the hold-out dataset. When testing the recursively reduced models, the balanced accuracy remained near constant up to the model with 9 remaining features (89.5% balanced accuracy with 87.4% sensitivity and 91.6% specificity). A further reduction (removing the least important feature from the set of 9) resulted in a 12.3 percent point reduction in balanced accuracy primarily due to misclassification of the not willing to vaccinate (Specificity = 55.6%).

The features coming out of our model (Fig 3) revealed five main indicators: belief whether the COVID-19 vaccine would be safe and effective, belief whether a COVID-19 vaccine would

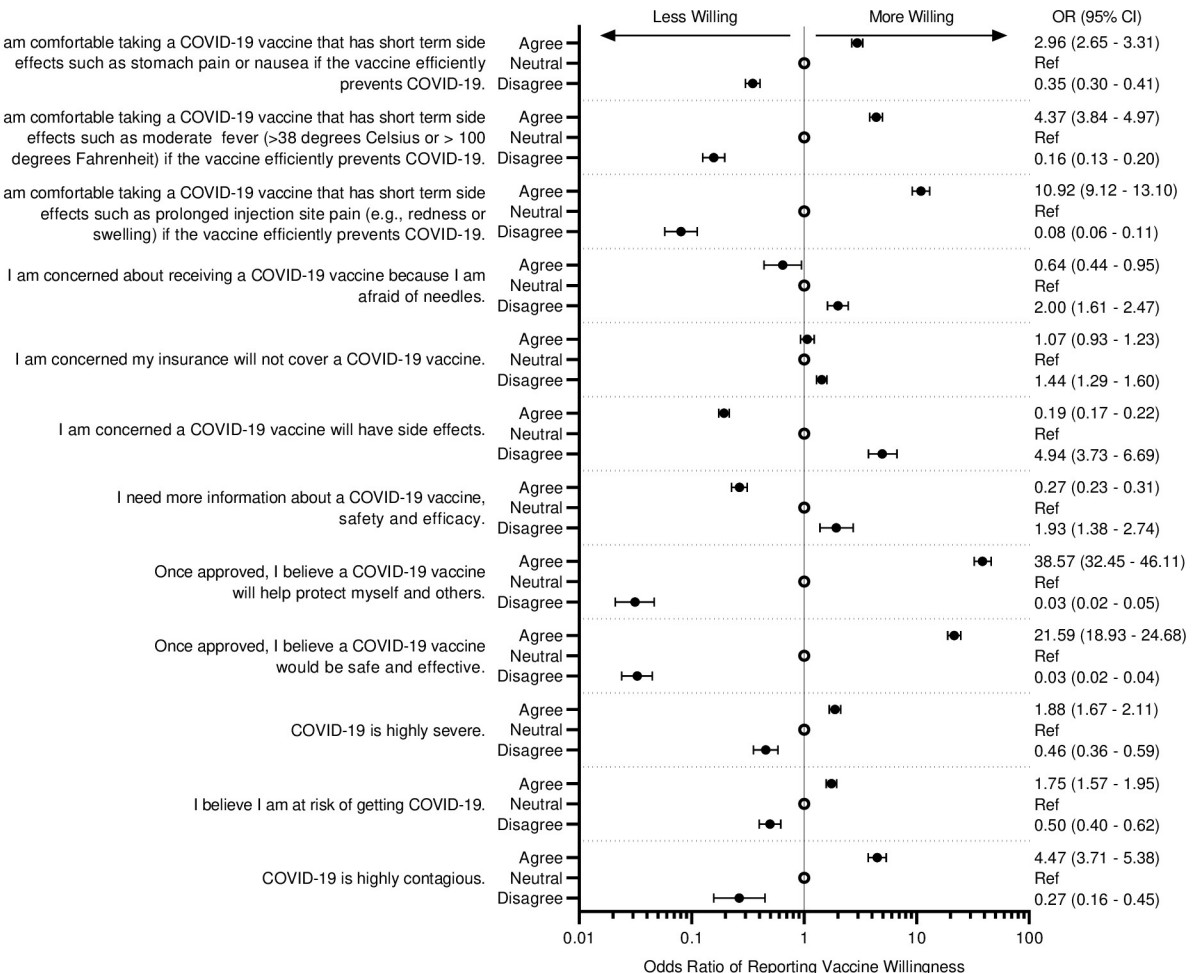

**Fig 2. Forest plot of willingness to vaccinate by survey response.** Shown are Odds Ratios (95% CI) for willingness to vaccinate for the different demographic characteristics. Odds Ratios were calculated using ordered logistic regression model with the 4 levels of willingness to be vaccinated as the outcome while adjusting for gender and race. Reference for each survey question is the option 'neutral' and is indicated by an open circle. For example subjects who agreed with the question '*I am comfortable taking a COVID-19 vaccine that has short term side effects such as stomach pain or nausea if the vaccine efficiently prevents COVID-19.*' are 3.0 times more likely to be more willing as compared to those who selected 'neutral'.

help protect "myself and others," degree of comfort with potential short term side effects from a COVID-19 vaccine, the belief whether vaccines in general are safe and important, and whether the respondent was Black or African American.

## Impact of positive vaccine news

Willingness to vaccinate increased after November 9th when results of the Pfizer Phase 3 vaccine trial were released (odds ratio 1.41, 95% CI 1.21–1.65; S4 Table). Prior to that date, 2.8% of adults were not at all willing to receive a COVID-19 vaccine and 6.3% were "not very" willing; after this these decreased to 1.4% and 5.9%, respectively. No meaningful difference was found in gender, age, BMI, education, income or race between the population who completed the survey before the Pfizer announcement on November 9th and those completing the survey after, between November 10th and 15th.

**Table 2. Talking to Healthcare provider within levels of willing to vaccinate.**

| Race | Gender | Variable | Not at all willing | Not very willing | Somewhat willing | Very willing |
|---|---|---|---|---|---|---|
| Asian | Women | Talk with HCP (No) | 0 (0.0%) | 5 (45.5%) | 9 (22.0%) | 1 (2.0%) |
| | | Talk with HCP (Yes) | 1 (100%) | 6 (54.5%) | 32 (78.0%) | 48 (98.0%) |
| | Men | Talk with HCP (No) | 3 (100%) | 5 (83.3%) | 7 (18.9%) | 17 (18.9%) |
| | | Talk with HCP (Yes) | 0 (0.0%) | 1 (16.7%) | 30 (81.1%) | 73 (81.1%) |
| Black or African American | Women | Talk with HCP (No) | 6 (46.2%) | 2 (9.1%) | 2 (4.2%) | 1 (3.6%) |
| | | Talk with HCP (Yes) | 7 (53.8%) | 20 (90.9%) | 46 (95.8%) | 27 (96.4%) |
| | Men | Talk with HCP (No) | 1 (50.0%) | 3 (25.0%) | 2 (5.7%) | 3 (13.0%) |
| | | Talk with HCP (Yes) | 1 (50.0%) | 9 (75.0%) | 33 (94.3%) | 20 (87.0%) |
| White | Women | Talk with HCP (No) | 56 (48.7%) | 56 (23.8%) | 83 (8.2%) | 157 (9.0%) |
| | | Talk with HCP (Yes) | 59 (51.3%) | 179 (76.2%) | 930 (91.8%) | 1596 (91.0%) |
| | Men | Talk with HCP (No) | 31 (58.5%) | 38 (27.0%) | 72 (8.7%) | 299 (11.3%) |
| | | Talk with HCP (Yes) | 22 (41.5%) | 103 (73.0%) | 755 (91.3%) | 2358 (88.7%) |

An overview of those who would talk to their Healthcare Provider (*'I would talk to my healthcare provider when considering a COVID-19 vaccine, before deciding whether or not to receive the vaccine'*) among the 4 levels of willingness, by race and gender. Note: due to small numbers, Native Hawaiian or Other Pacific Islander (n = 8), American Indian or Alaska Native (n = 21), Two or more (n = 44) and Prefer not to Answer (n = 114) were excluded from the table.

Notably, Moderna also released positive preliminary data regarding their vaccine on November 16[th] [20]. By this date 95% (7065 out of the 7,402) of the respondents had already completed the survey. To test if a similar effect of the Moderna news was present, surveys completed after the Pfizer announcement (November 9) and before the Moderna announcement (November 16) were compared to those completed after that announcement (November 17–20). A similar effect was observed (odds ratio of 1.27, 95%CI 0.91–1.80), though not statistically significant due to small sample sizes.

## Discussion

The recent approval of vaccines for COVID-19 has increased the focus on the need to maximize public vaccine acceptance. Vaccine uptake in older adults is of particular importance as increasing age is the leading risk factor for mortality and complications from COVID-19 infections [15, 16]. In this large survey of adults age 65 and older conducted in the United States in November 2020, the vast majority (91%) of adults reported that they are willing to receive a COVID-19 vaccine. However, even in this population of people with sufficient health literacy and trust in the healthcare system to participate in a digital clinical trial, 1 in 11 reported unwillingness to receive a vaccine. More concerningly, rates of vaccine unwillingness were higher in Black or African American adults and those at lower income and education levels, suggesting the potential for uneven vaccine uptake in at-risk communities.

In this study, Black or African American race was the only factor associated with willingness to immunize after accounting for beliefs about the vaccine; 1 in 4 Black or African American participants reported unwillingness to receive a vaccine. This is slightly lower than previous reports of ~40% or more unwilling [8, 9, 11, 23, 24], but is consistent with other studies that show that willingness among Black or African Americans is improving [10]. We also found lower rates of reported vaccine willingness in women compared with men, a factor that has been shown in other studies [8, 9, 11, 14, 23]. Socioeconomic factors, including higher education and higher income, were also associated with increased reported willingness to be vaccinated, similar to other studies [9, 11]. Notably, these findings contrast with characteristics of adults who refuse vaccines for their children. In studies of vaccine exemptions, higher income

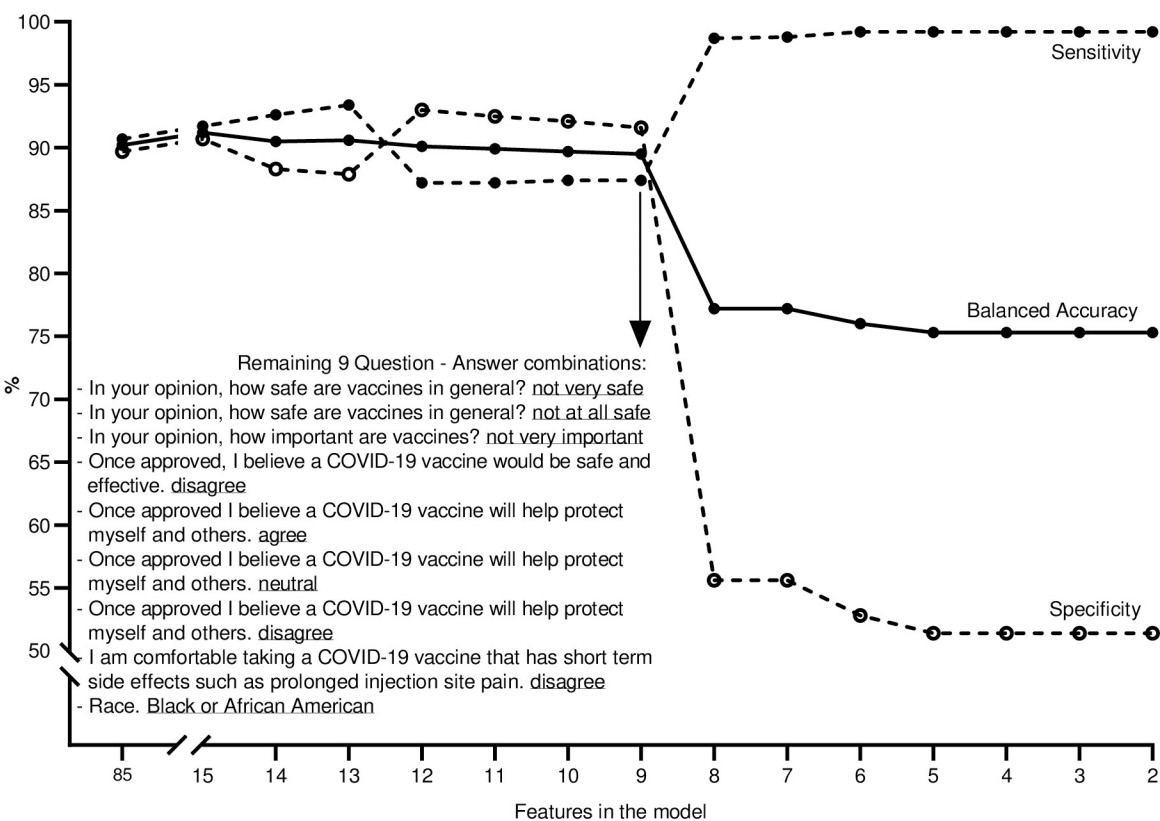

**Fig 3. Result of recursive feature elimination algorithms.** Random Forest classification algorithm was constructed to identify a set of determinants able to separate those who are not willing to vaccinate from those who are. The model started with a list of 85 features and predicted the willingness of subjects in the hold-out dataset with 90.2% balanced accuracy (solid line), which is an average of 90.7% sensitivity (dashed line closed circles) and 89.7% specificity (dashed line open circles). The balanced accuracy remained near constant when testing the recursively reduced models, up to the model with 9 remaining features (i.e. 5 questions with a total of 9 answers, see inserted table). This final model showed an 89.5% balanced accuracy with 87.4% sensitivity and 91.6% specificity. Further reduction, removing the least important feature from the set of 9 (i.e. 'Neutral' to *'Once approved, I believe a COVID-19 vaccine will help protect myself and others'*), resulted in a 12.3 percent point reduction in balanced accuracy primarily due to misclassification of the not willing to vaccinate (Specificity = 55.6%).

and education are often associated with higher rates of vaccine refusal [25]. Given that lower income communities and communities of color are at higher risk of COVID-19, vaccine hesitancy in these groups is of particular concern. Even if overall vaccine uptake is high, clustering of unimmunized persons can lead to continued circulation of vaccine-preventable diseases [26].

The strongest factors associated with and indicative of vaccine willingness in this population were beliefs about the COVID-19 vaccine's safety and efficacy, and a more altruistic belief whether a COVID-19 vaccine will help protect "myself and others." Vaccine safety and efficacy are among the most important factors reported influencing likeliness to vaccinate among Americans [9, 27] and Americans 65 and over [11]. Among those disinclined to vaccinate, the majority stated that they would discuss their decision with their healthcare provider, providing an important opportunity for education. Despite differences in rates of vaccine willingness by gender and race, the associations between beliefs about a COVID-19 vaccine and reported willingness to receive a vaccine were consistent across subgroups. In addition to broader public education campaigns about COVID-19, specific efforts should be made to facilitate healthcare patient-provider communication about COVID-19 vaccine focused on vaccine safety and efficacy.

While the study was not designed to determine the impact of vaccine-related news, the release of results from Pfizer's Phase 3 vaccine trial occurring in the middle of the survey [19] provided an opportunity to evaluate the impact of positive vaccine news on vaccine attitudes. Increased vaccine willingness was seen after the trial results were released, suggesting that public willingness to vaccinate may continue to rise as additional positive data are released. These data are supported by a survey experiment demonstrating those who were shown hypothetical messaging about the safety/efficacy of a vaccine were more likely, compared to a control group, to report they would take the vaccine [27]. On the other hand, given the potential for vaccine related data to shift perception, there remains the possibility that negative news stories about the vaccine (including reports of adverse events) may negatively impact vaccine willingness.

The high willingness among participants in our population to be vaccinated is similar to that reported in other assessments in older adults, though higher than the general population. Recently reported results from the Kaiser Family Foundation found that 85% of those 65 and older were willing to receive a vaccine [10]. Older adults consistently express a higher willingness to accept a COVID-19 vaccine than younger populations [3, 5, 6, 8, 11, 14]. Reasons for this may include a higher perceived risk of COVID-19 illness compared to younger adults [28], prior experience with vaccine preventable diseases and mass vaccination campaigns for diseases such as polio, or comfort with routine immunizations due to being recommended for influenza vaccine. In contrast, increasing age (over 50 or 60) does not appear to be a significant factor to predict willingness to engage in other preventive measures for COVID-19 including willingness to isolate [29]. In fact, certain behaviors known to be effective against COVID-19 spread, like mask wearing, were actually negatively correlated with age in that study (through age 80 years) [29]. In addition to the reasons mentioned above, other possible explanations for higher willingness to vaccinate in our study could include a high level of trust of healthcare providers, who are generally strongly pro-vaccination, and a selection bias from those willing and eligible to participate in the Heartline™ Study towards more openness to vaccination.

This study demonstrates the power of the digital platform used in the Heartline™ Study to rapidly generate real-world data. There are key features of this platform that enable rapid data generation: a properly constructed informed consent form that permits ad hoc survey deployment for data collection; a mobile app capable of pushing content and gathering variable but structured data; a back-end data structure that enables rapid analysis; and an app design that keeps participants highly engaged throughout the entire study, as evidenced by 83.6% of participants who took the survey. This platform opens the possibility of studies that investigate multiple effects over time or serial interventions in a population.

These findings should be interpreted in the context of several potential limitations and the implications of this study's findings should be understood in the context of the particular population recruited into the Heartline™ Study. There is an inherent bias that comes with studying populations who have chosen to enroll into a clinical study and where recruitment is done through a mobile App platform. Bias could have been introduced in Heartline™, such as selection of those with access to iPhones, which are premium mobile hardware, those familiar with app-based technologies, and those who were willing to share extensive amounts of their health information digitally. In addition, those with interest in mobile applications related to their health may be healthier and/or more health-conscious, and therefore may be more likely to vaccinate. Heartline™ participants have less representation of Black or African Americans (2.5% in this study vs 9% of 65 and older U.S. population) [30] and Asians (3.2% in this study vs 4% of 65 and older U.S population) [30] compared to the general U.S. 65+ population and skew higher in education and income, which may have led to an overestimate of vaccine acceptance compared with the general population. Nevertheless, finding that 1 in 11 patients in this

selected group are unwilling to be vaccinated may portend even lower rates in a broader population. Next, our analysis of the impact of vaccine-related news occurred during a time period when infections continued to rise; whether changes in attitudes were due to vaccine news or due to other factors is unclear. We also can not rule out that differences in vaccine willingness may be due to differences in the populations who completed the surveys at different time periods, though we are reassured that the demographic characteristics of the population did not appear to differ temporally. Finally, we asked about vaccine willingness, which may not directly translate into behavior, particularly if attitudes shift over time. We also highlight several strengths, including a high response rate (>80%), a short period of data collection (2 weeks), and the ability to deploy the survey rapidly in response to the pandemic.

In a 65 and older U.S. population, most are willing to be vaccinated, with Black or African American participants and females significantly less willing. The majority would be willing to discuss their concerns with their providers, who could leverage the beliefs identified here in tailoring a message to encourage vaccination. Potential health care policy implications from these findings include using providers as a key leverage point and ensure appropriate resources for them to reach and educate this older, at risk population to address vaccine hesitancy. In addition, policies to start vaccinations with older populations first could result in faster adoption and make vaccination more acceptable to those who prefer to 'wait and see' [7]. Developing implementable strategies to consistently communicate the potential benefits of vaccination, including their safety and efficacy, could improve acceptance and help speed the efforts to thwart this global pandemic.

## Supporting information

**S1 Fig. Forest plot of willingness to vaccinate by survey response (general vaccine beliefs).** Shown are Odds Ratios (95% CI) for willingness to vaccinate. Odds Ratios were calculated using ordered logistic regression model with the 4 levels of willingness to be vaccinated as the outcome while adjusting for gender and race.
(PDF)

**S2 Fig. Forest plots of willingness to vaccinate by COVID-19 beliefs and race.** Shown are Odds Ratios (95% CI) for willingness to vaccinate for the survey by race: White (A), Asian (B), and Black (C). Odds Ratios were calculated using ordered logistic regression model with the 4 levels of willingness to be vaccinated as the outcome while adjusting for gender and race. Reference for each survey question is the option 'neutral' and is indicated by an open circle. *na* indicates insufficient subjects for this category.
(PDF)

**S1 Table. Vaccine hesitancy assessment.**
(DOCX)

**S2 Table. Recursive feature elimination algorithm feature definitions.**
(DOCX)

**S3 Table. Willingness to be vaccinated by model features from survey.**
(DOCX)

**S4 Table. Impact of positive vaccine efficacy news by level of willing to vaccinate.** November 9, 2020, Pfizer announced a first interim analysis reporting >90% efficacy of their vaccine candidate. The impact of this news on willingness was evaluated among those who answered the survey before the 9th to those who answered thereafter (excluding those who answered on the 9th itself). Those after November 9th were more willing to vaccinate (odds ratio 1.41, 95%

CI 1.21–1.65).
(DOCX)

## Acknowledgments

The investigators would like to thank all of the participants in the Heartline™ study for their partnership in this research.

## Author Contributions

**Conceptualization:** Janeta Nikolovski, Martin Koldijk, Gerrit Jan Weverling, Paul Burton.

**Data curation:** Janeta Nikolovski, Martin Koldijk, Gerrit Jan Weverling, Robert Zambon, Nnamdi Ezeanochie, Jennifer Turgiss, Robyn Jones, Paul Burton.

**Formal analysis:** Janeta Nikolovski, Martin Koldijk, Gerrit Jan Weverling.

**Investigation:** Janeta Nikolovski, Martin Koldijk, Gerrit Jan Weverling, John Spertus, Mintu Turakhia, Leslie Saxon, Mike Gibson, John Whang, Troy Sarich, Robert Zambon, Robyn Jones, Paul Burton, Ann Marie Navar.

**Methodology:** Janeta Nikolovski, Martin Koldijk, Gerrit Jan Weverling, John Spertus, John Whang, Robert Zambon, Nnamdi Ezeanochie, Jennifer Turgiss, Robyn Jones, Paul Burton.

**Supervision:** Janeta Nikolovski, Gerrit Jan Weverling, John Whang, Troy Sarich, Paul Burton, Ann Marie Navar.

**Visualization:** Martin Koldijk.

**Writing – original draft:** Janeta Nikolovski, Martin Koldijk, Gerrit Jan Weverling.

**Writing – review & editing:** Janeta Nikolovski, Martin Koldijk, Gerrit Jan Weverling, John Spertus, Mintu Turakhia, Leslie Saxon, Mike Gibson, John Whang, Troy Sarich, Robert Zambon, Nnamdi Ezeanochie, Jennifer Turgiss, Robyn Jones, Jeff Stoddard, Paul Burton, Ann Marie Navar.

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
