## [Decision Letter · Decision Letter 0]

17 Feb 2021

PONE-D-21-00518

Factors indicating intention to vaccinate with a COVID-19 vaccine among older U.S. Adults

PLOS ONE

Dear Dr. Navar,

Thank you for submitting your manuscript to PLOS ONE. After careful consideration, we feel that it has merit but does not fully meet PLOS ONE’s publication criteria as it currently stands. Therefore, we invite you to submit a revised version of the manuscript that addresses the points raised during the review process.

Both reviewers agree that the paper is well written and addresses a relevant and timely question. However, they also raise a number of concerns and provide comments and suggestions that, if properly addressed, will help to further improve the quality of the paper. I also outline my comments below. 

First, both reviewers believe that more should be done to connect your research with the literature. They provide useful references, and you will certainly find recent work very relevant to the topic. Moreover, R2 is less satisfied than R1 regarding the importance of the research, and I encourage you to address this consideration. One way to do it could be to provide more information about the “high risk” that you refer to on page 4, which increases the relevance of improving our knowledge about elderly people and their responses to the crisis.

Second, and it is a crucial point: the reviewers raise a number of concerns with the timing. I see three issues. As mentioned by R1, you should address the news around the Moderna’s vaccine. R2 details some concerns about the representativeness of the sample in regard of the ‘treatment’ (i.e pre/post Pfizer news). Finally, I would also like you to address the possibility that the treatment (or the effect of the news) might have potential heterogenous effects across subsamples (for examples, groups that are less likely to trust scientists). 

Third, on top of the representativeness in the pre/post-Pfizer news samples, both reviewers raise fair questions about the whole sample with clear comments related to respondents’ income, the opt-in panel from an iPhone app, the external validity, etc. 

See also the other concerns outlined quite clearly by the reviewers below. 

We look forward to receiving your revised manuscript.

Kind regards,

Jean-François Daoust

Academic Editor

PLOS ONE

Journal Requirements:

2.Thank you for including your ethics statement: 

"clinicaltrials.gov, NCT04276441

Approved by WIRB. Study number 1260314. IRB Tracking Number 20191385.

All study participants provided informed consent to analyze de-identified data, including surveys delivered through the HeartlineTM app. ".   

b)Please provide additional details regarding participant consent. In the ethics statement in the Methods and online submission information, please ensure that you have specified what type you obtained (for instance, written or verbal, and if verbal, how it was documented and witnessed). If your study included minors, state whether you obtained consent from parents or guardians. If the need for consent was waived by the ethics committee, please include this information.

3.We note that you have indicated that data from this study are available upon request. PLOS only allows data to be available upon request if there are legal or ethical restrictions on sharing data publicly. For information on unacceptable data access restrictions, please see http://journals.plos.org/plosone/s/data-availability#loc-unacceptable-data-access-restrictions.

5.Thank you for stating the following in the Competing Interests section:

"I have read the journal's policy and the authors of this manuscript have the following competing interests:

JN, MK, GJW, JW, TS, JS, PB, RZ, NE, JT, RJ are employees of Janssen and Johnson and Johnson.

JS reports personal fees from Amgen, personal fees from Bayer, personal fees from Merck, personal fees from Novartis, personal fees from Janssen, personal fees from Myokardia, personal fees from Blue Cross Blue Shield of Kansas City,  outside the submitted work;  In addition, Dr. Spertus has a patent Copyright to the KCCQ with royalties paid and Equity in Health Outcomes Sciences.

MT reports grants from Janssen Inc, personal fees from Medtronic Inc, personal fees from Abbott, grants from Boehringer lngelheim, grants and personal fees from Cardiva Medical, personal fees from iRhythm, grants from Bristol Myers Squibb, grants from American Heart Association, grants from SentreHeart, personal fees from Novartis, personal fees from Biotronik, personal fees from Sanofi, personal fees from Pfizer, grants from Apple, grants from Bayer, personal fees from Myokardia, personal fees from Johnson & Johnson, personal fees from Milestone Pharmaceuticals, outside the submitted work; and Dr. Turakhia is an editor for JAMA Cardiology.

LS reports personal fees from J&J, outside the submitted work.

MG receives research grant support from Janssen.

AMN receives consulting fees from Janssen for serving on the steering committee of the Heartline study.  In addition, Dr. Navar receives funding for research to her institution from Amgen and Janssen, and honoraria and consulting fees from Amarin, Amgen, Astra Zeneca, BI, Esperion, Lilly, Sanofi, Regeneron, NovoNordisk, Novartis, The Medicines Company, New Amsterdam, Cerner, 89Bio, and Pfizer, outside the scope of this work."

Reviewers' comments:

Reviewer's Responses to Questions

**Comments to the Author**

1. Is the manuscript technically sound, and do the data support the conclusions?

Reviewer #1: Partly

Reviewer #2: Yes

2. Has the statistical analysis been performed appropriately and rigorously? 

Reviewer #1: Yes

Reviewer #2: Yes

3. Have the authors made all data underlying the findings in their manuscript fully available?

Reviewer #1: No

Reviewer #2: Yes

4. Is the manuscript presented in an intelligible fashion and written in standard English?

Reviewer #1: Yes

Reviewer #2: Yes

5. Review Comments to the Author

Reviewer #1: This piece reviews the demographic and attitudinal correlates of intentions to refuse vaccination for COVID-19, among older US adults (aged 65+) . The authors find that Black respondents and women are significantly more likely to intend to refuse vaccination than the 65+ population more generally. The authors also find that concerns about vaccine safety and efficacy were correlated with vaccination intentions, that most individuals who intend to refuse vaccination would nevertheless be willing to consult with a medical professional prior to doing so, and that intentions to vaccinate significantly increased following news of Pfizer's positive Stage 3 clinical results.

In general, I found this paper to be well written, substantively important, and methodologically rigorous. However, while I find the piece suitable for publication in a journal like this one, there are several areas where I believe the authors can offer additional conceptual and analytical detail. I have listed each of those concerns in the order in which I encountered each one in the manuscript. I also suggest several actions the authors might take to remedy those concerns, when revising this manuscript.

1. p. 4 (final paragraph): The authors do an excellent job motivating why it is important to study vaccination intentions among older Americans -- i.e., because they are particularly at risk of experiencing severe complications from contracting COVID-19.

However, I think the authors could do much more to put their research into dialogue with other work on the demographic correlates of COVID-19 vaccine refusal, as well as individuals' reasons for planning to refuse to vaccinate. For example, Callaghan et al (2021) report a nearly identical pattern of demographic correlates of vaccine hesitancy (i.e., that women and Black respondents are less likely to intend to vaccinate) as well as individuals' reasons for intending to not vaccinate (e.g., concerns about safety and efficacy).

Why might we expect to observe a similar (or, different?) pattern of results among older Americans? What do we learn about either the nature of COVID-19 vaccine hesitancy, or its potential health policy implications, from studying this group? I encourage the authors to consider expanding on this point when discussing the importance and novelty of their research.

2. p. 7, (penultimate paragraph) and pp. 15-16 (final Results sub-section): While I found the results of the pre/post Pfizer Clinical Trial news to be both interesting and informative, I have two major conceptual and methodological concerns about this analysis.

A. First, the study's enrollment period (11/6-20, 2020) included not only the release of Pfizer's clinical trial data (11/9), but Moderna's as well (11/16). It was unclear to me why the authors analyze the former, but not the latter, when estimating the effects of attention to positive vaccine-related news. (see: https://www.nytimes.com/interactive/2020/science/coronavirus-vaccine-tracker.html)

If the authors have the ability to replicate this analysis for the Moderna news, I encourage them to do so. If not (e.g., due to statistical power concerns), I encourage them to instead state what those concerns are.

B. I am also somewhat concerned about the estimation of news "treatment" effects in the pre vs. post Pfizer announcement groups. Because this study is correlational, it could be the case that the group of respondents interviewed later on in the survey administration process was systematically different from the pre-treatment group on demographic on other factors associated with vaccine uptake. For example, if men were more likely to take the survey later on in the administration process than women, the appearance of a news treatment effect could actually be the result of methodological artifact: i.e., a gender effect.

Although the authors lack the ability to test these claims longitudinally (which would allow for the estimation of *within* subject treatment effects; potentially alleviating the concerns mentioned above), the authors could instead address this concern by showing that the pre and post treatment subgroups were balanced with respect to (1) demographic characteristics and (2) reasons provided for intending/not intending to vaccinate. This would help ensure that attention to the vaccine-related news, and not differences between the subsamples, is responsible for the effects documented in the piece.

3. p. 8 (Table 1, row 1): the authors observe much lower levels of vaccine hesitancy among older survey respondents than has been typically observed in existing research on COVID-19 vaccine hesitancy (e.g., see the Callaghan et al., piece referenced in #1). It is somewhat at odds with recent findings (e.g., Daoust 2020) suggesting that older and younger people tend to have similar attitudinal and behavioral responses to the pandemic.

If the authors had a priori expectations regarding why older individuals might be more likely to intend to vaccinate (e.g., increased risk of experiencing severe health complications), I encourage them to expand on that rationale. If not, I invite them to speculate, perhaps in the piece's discussion section, as to why this might be the case. Either way, I think the authors should make more of an effort to put these findings into conversation with types of research cited above; i.e., to note that older folks may be exceptional in their intentions to vaccinate, and that the effect of age on vaccination intentions may differ from the effects of age in other attitudinally and behaviorally relevant domains.

4. p. 19: the authors do an excellent job recognizing the limitations of collecting responses from an opt-in sample of mobile health app users. In addition to the concerns listed already in the document, I encourage the authors to discuss potential limits of the mobile app platform *itself* as a recruitment mechanism for this study. In addition to demographic differences in mobile phone access and representation in clinical trial enrollment, individuals who choose to download and regularly consult phone applications related to their health may be healthier and/or more health-conscious than individuals who do not.

Consequently, while observing higher-than-typical rates of vaccine compliance among older individuals could reflect an age cohort effect (see: point #3 above), it could alternatively reflect differences in who chooses to use the platform (vs. the remainder of the 65+ population). I.e., it could be the case individuals enrolled in this study and who regularly consult the app are more concerned about their health, and therefore more likely to vaccinate; or that they are more attuned to the possibility of experiencing adverse vaccination effects?

I encourage the authors to expand on this point in the piece's discussion. If possible, comparing the demographic characteristics of the 65+ sample to known subpopulation demographic benchmarks (e.g., from the US Census) would help further underscore the representativeness of this sample.

Minor points:

- The authors refer to study participants as "Adults" in the piece's abstract. I think it would be more informative to list the study's target population here; i.e., US Adults aged 65 or older.

- On p.5, the authors note that the study was approved by "an IRB." Which IRB, specifically, approved this research? I encourage the authors to provide additional information here.

References

Callaghan, Timothy, Ali Moghtaderi, Jennifer A. Lueck, Peter J. Hotez, Ulrich Strych, Avi Dor, Erika Franklin Fowler, and Matt Motta. "Correlates and disparities of COVID-19 vaccine hesitancy." Available at SSRN 3667971 (2020).

Daoust, J. F. (2020). Elderly people and responses to COVID-19 in 27 Countries. PloS one, 15(7), e0235590.

Reviewer #2: Thank you for the opportunity to review "Factors Indicating Intention to Vaccinate with a COVID-19 Vaccine among Older U.S. Adults" for possible publication in PLOS One. This article relies on an original survey sample of adults enrolled in the Heartline clinical study to examine vaccination intentions among U.S. adults over the age of 65. The authors find several factors influence the decision to vaccinate, include race, gender, and beliefs about the safety and efficacy of COVID-19. While this paper is written on a timely and vital topic, a few issues need to be resolved before publication.

My first concern with the manuscript in its current form is the generalizability of the findings. I am concerned the findings shown here may not be representative of the U.S. population. For example, examining Table 1 shows that 92% of the sample identifies as White and over 50% of the survey has income over $75,000. While I believe it is important to study the vaccine intentions of this vulnerable population, I don't believe the sample the authors use are representative of the older population in the U.S. The authors do make reference to the limitations of their study in the discussion but I believe the authors need to be more cautious about the inferences readers should take from the article.

My second concern with the manuscript is I would have liked to see more literature on vaccine hesitancy, specifically related to COVID-19, given all the recent literature done on the issue. I would encourage the authors to look at articles from Callaghan et al.; Motta et al. (just to name a few). I also would have liked to see more framing up front about the importance of studying this vulnerable population.

Thank you again for the opportunity to review this research letter. With these key changes, I believe this manuscript will make a valuable addition to our understanding of the factors influencing individual decisions to vaccinate against COVID-19.

6. PLOS authors have the option to publish the peer review history of their article (what does this mean?). If published, this will include your full peer review and any attached files.

Reviewer #1: **Yes: **Matt Motta

Reviewer #2: No

---

## [Author Response · Author response to Decision Letter 0]

8 Apr 2021

Reviewer #1: This piece reviews the demographic and attitudinal correlates of intentions to refuse vaccination for COVID-19, among older US adults (aged 65+) . The authors find that Black respondents and women are significantly more likely to intend to refuse vaccination than the 65+ population more generally. The authors also find that concerns about vaccine safety and efficacy were correlated with vaccination intentions, that most individuals who intend to refuse vaccination would nevertheless be willing to consult with a medical professional prior to doing so, and that intentions to vaccinate significantly increased following news of Pfizer's positive Stage 3 clinical results.

In general, I found this paper to be well written, substantively important, and methodologically rigorous. However, while I find the piece suitable for publication in a journal like this one, there are several areas where I believe the authors can offer additional conceptual and analytical detail. I have listed each of those concerns in the order in which I encountered each one in the manuscript. I also suggest several actions the authors might take to remedy those concerns, when revising this manuscript.

1. p. 4 (final paragraph): The authors do an excellent job motivating why it is important to study vaccination intentions among older Americans -- i.e., because they are particularly at risk of experiencing severe complications from contracting COVID-19.

However, I think the authors could do much more to put their research into dialogue with other work on the demographic correlates of COVID-19 vaccine refusal, as well as individuals' reasons for planning to refuse to vaccinate. For example, Callaghan et al (2021) report a nearly identical pattern of demographic correlates of vaccine hesitancy (i.e., that women and Black respondents are less likely to intend to vaccinate) as well as individuals' reasons for intending to not vaccinate (e.g., concerns about safety and efficacy).

We greatly appreciate the specific reference and agree with the reviewer. Callaghan et al (2021) indeed supports our findings nicely and has now been added in several places in our manuscript. Additional references have also been added to further support our discussion regarding demographic correlates of vaccine hesitancy.

p.18: “Vaccine safety and efficacy are among the most important factors reported influencing likeliness to vaccinate among Americans (Callaghan, Palm) and Americans 65 and over (Malani).”

Why might we expect to observe a similar (or, different?) pattern of results among older Americans? What do we learn about either the nature of COVID-19 vaccine hesitancy, or its potential health policy implications, from studying this group? I encourage the authors to consider expanding on this point when discussing the importance and novelty of their research.

While we weren’t surprised to see lower levels of vaccine hesitancy in older Americans as others have reported similar findings, one of our main findings was that this older population trusts healthcare providers, which is also consistent with prior work (https://www.pewresearch.org/science/2019/08/02/findings-at-a-glance-medical-doctors/). We found that providers are the predominant source of information about vaccines for this population (vs. the internet or social media). The primary health policy implication is that providers are a key leverage point and need to be appropriately resourced to reach and educate this older, at risk population to address vaccine hesitancy. In addition, policies to start vaccinations with older populations first could result in faster adoption and make it more acceptable to those who prefer to ‘wait and see how it goes for others first’ (Talev Axios-Ipsos poll).

We have added further discussion regarding the nature of vaccine hesitancy and health policy implications in the discussion. 

p.18: “Vaccine safety and efficacy are among the most important factors reported influencing likeliness to vaccinate among Americans (Callaghan, Palm) and Americans 65 and over (Malani).”

p.21: “Potential health care policy implications from these findings include using providers as a key leverage point and ensure appropriate resources for them to reach and educate this older, at risk population to address vaccine hesitancy. In addition, policies to start vaccinations with older populations first could result in faster adoption and make vaccination more acceptable to those who prefer to ‘wait and see’ [Talev].”

2. p. 7, (penultimate paragraph) and pp. 15-16 (final Results sub-section): While I found the results of the pre/post Pfizer Clinical Trial news to be both interesting and informative, I have two major conceptual and methodological concerns about this analysis.

A. First, the study's enrollment period (11/6-20, 2020) included not only the release of Pfizer's clinical trial data (11/9), but Moderna's as well (11/16). It was unclear to me why the authors analyze the former, but not the latter, when estimating the effects of attention to positive vaccine-related news. (see: https://www.nytimes.com/interactive/2020/science/coronavirus-vaccine-tracker.html)

If the authors have the ability to replicate this analysis for the Moderna news, I encourage them to do so. If not (e.g., due to statistical power concerns), I encourage them to instead state what those concerns are.

 The reviewer is correct: the reason this was not in our initial analysis was due to lack of statistical power. Moderna released their first preliminary data on November 16th, during the tail end of our survey. By then, we already had 95% of participants complete the survey, with only 337 completions at and after the 16th. To prevent contamination of that second positive news story on the inference regarding the Pfizer timing, we have revised our analysis to exclude the small number of surveys that were completed after the Moderna news was released on November 16th.

We now compare surveys completed before the Pfizer announcement (November 6-8) to those completed after that announcement (November 10-15), excluding any that came in after the Moderna announcement on the 16th. This analysis maintained a significant odds ratio of 1.41 (1.21-1.65). The original analysis gave an odds ratio of 1.46 (1.28-1.67). This has been updated in the Results on p.16. 

While underpowered, we did perform an exploratory analysis regarding the Moderna timing. To test if a similar effect of the Moderna news was present, surveys completed after the Pfizer announcement (November 9) and before the Moderna announcement (November 16) were compared to those completed after the Moderna announcement (November 17-20). A similar effect was observed (odds ratio of 1.27, 95%CI 0.91 – 1.80), though not statistically significant due to small sample sizes.

We have added the below comment to the manuscript on p.7-8 and have updated Table S4 with the new analysis.

p.7-8: “Shortly after the November survey was offered, Pfizer announced a first interim analysis reporting >90% efficacy of their vaccine candidate on November 9, 2020 [16]. We investigated if that news had an impact on willingness to vaccinate in our population and compared those who answered the survey before the 9th to those who answered after (excluding those who answered on the 9th itself, and those who answered on or after November 16th, the Moderna announcement) using an ordered logistic regression model with four-level willingness to be vaccinated as the outcome and an indicator variable for the time period (before and after November 9th) and gender as predictors.”

B. I am also somewhat concerned about the estimation of news "treatment" effects in the pre vs. post Pfizer announcement groups. Because this study is correlational, it could be the case that the group of respondents interviewed later on in the survey administration process was systematically different from the pre-treatment group on demographic on other factors associated with vaccine uptake. For example, if men were more likely to take the survey later on in the administration process than women, the appearance of a news treatment effect could actually be the result of methodological artifact: i.e., a gender effect.

Although the authors lack the ability to test these claims longitudinally (which would allow for the estimation of *within* subject treatment effects; potentially alleviating the concerns mentioned above), the authors could instead address this concern by showing that the pre and post treatment subgroups were balanced with respect to (1) demographic characteristics and (2) reasons provided for intending/not intending to vaccinate. This would help ensure that attention to the vaccine-related news, and not differences between the subsamples, is responsible for the effects documented in the piece.

The reviewer makes good points and we investigated further. We compared gender, age, BMI, education, income and race in the populations who completed the survey prior to Pfizer’s announcement to those completing after (but prior to Moderna’s announcement). We found no meaningful difference in any of those characteristics between the two populations. 

We did not evaluate reasons for intention to vaccinate or trust in providers before in after the news as we feel these belief variables may be impacted by the positive news itself. 

 The text below was added to the Results on p.16.

p.16: “No meaningful difference was found in gender, age, BMI, education, income or race between the population who completed the survey before the Pfizer announcement on November 9th and those completing the survey after, between November 10th and 15th. Notably, Moderna also released positive preliminary data regarding their vaccine on November 16th[ref]. By this date 95% (7065 out of the 7,402) of the respondents had already completed the survey. To test if a similar effect of the Moderna news was present, surveys completed after the Pfizer announcement (November 9) and before the Moderna announcement (November 16) were compared to those completed after that announcement (November 17-20). A similar effect was observed (odds ratio of 1.27, 95%CI 0.91 – 1.80), though not statistically significant due to small sample sizes.”

We have also recognized this as a limitation: 

 p.20-21: “We also cannot rule out that differences in vaccine willingness may be due to differences in the populations who completed the surveys at different time periods, though we are reassured that the demographic characteristics of the population did not appear to differ temporally.”

3. p. 8 (Table 1, row 1): the authors observe much lower levels of vaccine hesitancy among older survey respondents than has been typically observed in existing research on COVID-19 vaccine hesitancy (e.g., see the Callaghan et al., piece referenced in #1). It is somewhat at odds with recent findings (e.g., Daoust 2020) suggesting that older and younger people tend to have similar attitudinal and behavioral responses to the pandemic.

If the authors had a priori expectations regarding why older individuals might be more likely to intend to vaccinate (e.g., increased risk of experiencing severe health complications), I encourage them to expand on that rationale. If not, I invite them to speculate, perhaps in the piece's discussion section, as to why this might be the case. Either way, I think the authors should make more of an effort to put these findings into conversation with types of research cited above; i.e., to note that older folks may be exceptional in their intentions to vaccinate, and that the effect of age on vaccination intentions may differ from the effects of age in other attitudinally and behaviorally relevant domains.

As we note in the discussion, our high rate of vaccine willingness was similar to what has been seen in older adults in other studies. We have added a comment to note that this is higher than the general population.

p.19: “The high willingness among participants in our population to be vaccinated is similar to that reported in other assessments in older adults, though higher than the general population.”

We appreciate the Daoust 2020 reference which reports that when it comes to attitudes and compliance to different COVID-19 preventative measures, like willingness to isolate, advanced age (over 50 or 60) does not appear to be a significant factor, and indeed older and younger people tend to have similar attitudinal and behavioral responses to the pandemic. This is interesting given that age is a significant factor in COVID-19 mortality and certain behaviors known to be effective against COVID-19 spread, like mask wearing, were actually negatively correlated with age in that publication (through age 80 years). This has been added to the manuscript in the Discussion, p.19.

p.19: “In contrast, increasing age (over 50 or 60) does not appear to be a significant factor to predict willingness to engage in other preventative measures for COVID-19 including willingness to isolate (Daoust). In fact, certain behaviors known to be effective against COVID-19 spread, like mask wearing, were actually negatively correlated with age in that study (through age 80 years) (Daoust). In addition to the reasons mentioned above, other possible explanations for higher willingness to vaccinate in our study could include a high level of trust of healthcare providers, who are generally strongly pro-vaccination, and a selection bias from those willing and eligible to participate in the HeartlineTM Study towards more openness to vaccination.”

4. p. 19: the authors do an excellent job recognizing the limitations of collecting responses from an opt-in sample of mobile health app users. In addition to the concerns listed already in the document, I encourage the authors to discuss potential limits of the mobile app platform *itself* as a recruitment mechanism for this study. In addition to demographic differences in mobile phone access and representation in clinical trial enrollment, individuals who choose to download and regularly consult phone applications related to their health may be healthier and/or more health-conscious than individuals who do not.

Consequently, while observing higher-than-typical rates of vaccine compliance among older individuals could reflect an age cohort effect (see: point #3 above), it could alternatively reflect differences in who chooses to use the platform (vs. the remainder of the 65+ population). I.e., it could be the case individuals enrolled in this study and who regularly consult the app are more concerned about their health, and therefore more likely to vaccinate; or that they are more attuned to the possibility of experiencing adverse vaccination effects?

I encourage the authors to expand on this point in the piece's discussion. If possible, comparing the demographic characteristics of the 65+ sample to known subpopulation demographic benchmarks (e.g., from the US Census) would help further underscore the representativeness of this sample.

We agree with the reviewer’s comments and have added more detail to the limitations discussed on p. 20. We agree, as in any recruitment that is channel-specific – in our case, via mobile technologies and apps – bias can be introduced, such as selection of those with access to iPhones, which are premium mobile hardware; further selection bias of those familiar with app-based technologies; and selection of those willing to share extensive amounts of their health information digitally. Indeed, our recruited population may be healthier and/or more health conscious and more likely to vaccinate.

p.20: “These findings should be interpreted in the context of several potential limitations and the implications of this study’s findings should be understood in the context of the particular population recruited into the HeartlineTM Study. There is an inherent bias that comes with studying populations who have chosen to enroll into a clinical study and where recruitment is done through a mobile App platform. Bias could have been introduced in HeartlineTM, such as selection of those with access to iPhones, which are premium mobile hardware, those familiar with app-based technologies, and those who were willing to share extensive amounts of their health information digitally. In addition, those with interest in mobile applications related to their health may be healthier and/or more health-conscious, and therefore may be more likely to vaccinate.”

We have also drawn attention to the representativeness of our population related to Race, by comparing to the general U.S. 65+ population, now noting on p.20 , “…Black or African Americans (2.5% in this study vs 9% of 65 and older U.S. population) and Asians (3.2% in this study vs 4% of 65 and older U.S population)…”

Minor points:

- The authors refer to study participants as "Adults" in the piece's abstract. I think it would be more informative to list the study's target population here; i.e., US Adults aged 65 or older.

 We agree and have made that change. “U.S. adults aged 65 and older enrolled in the HeartlineTM clinical study…”

- On p.5, the authors note that the study was approved by "an IRB." Which IRB, specifically, approved this research? I encourage the authors to provide additional information here.

We agree and have added that detail to the Methods, p.5: “The study protocol was approved by Western Institutional Review Board and all participants submit a signed informed consent to analyze de-identified data through the HeartlineTM app.”

References

- Callaghan, Timothy, Ali Moghtaderi, Jennifer A. Lueck, Peter J. Hotez, Ulrich Strych, Avi Dor, Erika Franklin Fowler, and Matt Motta. "Correlates and disparities of COVID-19 vaccine hesitancy." Available at SSRN 3667971 (2020).

- Daoust, J. F. (2020). Elderly people and responses to COVID-19 in 27 Countries. PloS one, 15(7), e0235590.

Reviewer #2: Thank you for the opportunity to review "Factors Indicating Intention to Vaccinate with a COVID-19 Vaccine among Older U.S. Adults" for possible publication in PLOS One. This article relies on an original survey sample of adults enrolled in the Heartline clinical study to examine vaccination intentions among U.S. adults over the age of 65. The authors find several factors influence the decision to vaccinate, include race, gender, and beliefs about the safety and efficacy of COVID-19. While this paper is written on a timely and vital topic, a few issues need to be resolved before publication.

My first concern with the manuscript in its current form is the generalizability of the findings. I am concerned the findings shown here may not be representative of the U.S. population. For example, examining Table 1 shows that 92% of the sample identifies as White and over 50% of the survey has income over $75,000. While I believe it is important to study the vaccine intentions of this vulnerable population, I don't believe the sample the authors use are representative of the older population in the U.S. The authors do make reference to the limitations of their study in the discussion but I believe the authors need to be more cautious about the inferences readers should take from the article.

We agree with the reviewer’s comments and have added language to qualify our findings better, now noting on p.20, “These findings should be interpreted in the context of several potential limitations and the implications of this study’s findings should be understood in the context of the particular population recruited into the HeartlineTM Study.” 

We added more detail to the limitations discussed on p.20. We agree, as in any recruitment that is channel-specific – in our case, via mobile technologies and apps – bias can be introduced, such as selection of those with access to iPhones, which are premium mobile hardware; further selection bias of those familiar with app-based technologies; and selection of those willing to share extensive amounts of their health information digitally. Indeed, our recruited population may be healthier and/or more health conscious and more likely to vaccinate.

p.20: “These findings should be interpreted in the context of several potential limitations and the implications of this study’s findings should be understood in the context of the particular population recruited into the HeartlineTM Study. There is an inherent bias that comes with studying populations who have chosen to enroll into a clinical study and where recruitment is done through a mobile App platform. Bias could have been introduced in HeartlineTM, such as selection of those with access to iPhones, which are premium mobile hardware, those familiar with app-based technologies, and those who were willing to share extensive amounts of their health information digitally. In addition, those with interest in mobile applications related to their health may be healthier and/or more health-conscious, and therefore may be more likely to vaccinate.”

We have also drawn attention to the representativeness of our population related to Race, by comparing to the general U.S. 65+ population, now noting on p.19 , “…Black or African Americans (2.5% in this study vs 9% of 65 and older U.S. population) and Asians (3.2% in this study vs 4% of 65 and older U.S population)…”

My second concern with the manuscript is I would have liked to see more literature on vaccine hesitancy, specifically related to COVID-19, given all the recent literature done on the issue. I would encourage the authors to look at articles from Callaghan et al.; Motta et al. (just to name a few). I also would have liked to see more framing up front about the importance of studying this vulnerable population.

We agree and have updated our literature search and added several new references to the manuscript, including those mentioned. 

p.19: “In contrast, increasing age (over 50 or 60) does not appear to be a significant factor to predict willingness to engage in other preventative measures for COVID-19 including willingness to isolate (Daoust). In fact, certain behaviors known to be effective against COVID-19 spread, like mask wearing, were actually negatively correlated with age in that study (through age 80 years) (Daoust). In addition to the reasons mentioned above, other possible explanations for higher willingness to vaccinate in our study could include a high level of trust of healthcare providers, who are generally strongly pro-vaccination, and a selection bias from those willing and eligible to participate in the HeartlineTM Study towards more openness to vaccination.”

p.17: “Vaccine safety and efficacy are among the most important factors reported influencing likeliness to vaccinate among Americans (Callaghan, Palm) and Americans 65 and over (Malani).”

p.18: “These data are supported by a survey experiment demonstrating those who were shown hypothetical messaging about the safety/efficacy of a vaccine were more likely, compared to a control group, to report they would take the vaccine (Palm).”

• Malani PN, Solway E, Kullgren JT. Older Adults’ Perspectives on a COVID-19 Vaccine. JAMA Health Forum. Published online December 23, 2020. doi:10.1001/jamahealthforum.2020.1539

• Daoust J-F (2020) Elderly people and responses to COVID-19 in 27 Countries. PLoS ONE 15(7): e0235590. https://doi.org/10.1371/journal.pone.0235590

• Palm R, Bolsen T, Kingsland JT. The effect of frames on COVID-19 vaccine hesitancy. January 6, 2021. https://doi.org/10.1101/2021.01.04.21249241

• Callaghan, Timothy, Ali Moghtaderi, Jennifer A. Lueck, Peter J. Hotez, Ulrich Strych, Avi Dor, Erika Franklin Fowler, and Matt Motta. "Correlates and disparities of COVID-19 vaccine hesitancy." Available at SSRN 3667971 (2020).

We also added more discussion in the Introduction about the importance of studying this vulnerable population.

p.4: “Vaccine uptake in older adults is of particular importance as increasing age is the leading risk factor for mortality and complications from COVID-19 infections [13,19]. Guidelines have been developed that identify priority populations for vaccination (CDC.gov), with recommendations including those older than 65 years.“

p.4: “We hypothesized that older adults in our sample population would have a high willingness to vaccinate and sought to understand factors associated with those less willing.”

Thank you again for the opportunity to review this research letter. With these key changes, I believe this manuscript will make a valuable addition to our understanding of the factors influencing individual decisions to vaccinate against COVID-19.

---

## [Decision Letter · Decision Letter 1]

7 May 2021

Factors indicating intention to vaccinate with a COVID-19 vaccine among older U.S. Adults

PONE-D-21-00518R1

Dear Dr. Navar,

We’re pleased to inform you that your manuscript has been judged scientifically suitable for publication and will be formally accepted for publication once it meets all outstanding technical requirements.

Kind regards,

Jean-François Daoust

Academic Editor

PLOS ONE

Additional Editor Comments (optional):

Congratulations!

Reviewers' comments:

Reviewer's Responses to Questions

**Comments to the Author**

1. If the authors have adequately addressed your comments raised in a previous round of review and you feel that this manuscript is now acceptable for publication, you may indicate that here to bypass the “Comments to the Author” section, enter your conflict of interest statement in the “Confidential to Editor” section, and submit your "Accept" recommendation.

Reviewer #1: All comments have been addressed

Reviewer #2: All comments have been addressed

2. Is the manuscript technically sound, and do the data support the conclusions?

Reviewer #1: Yes

Reviewer #2: Yes

3. Has the statistical analysis been performed appropriately and rigorously? 

Reviewer #1: Yes

Reviewer #2: Yes

4. Have the authors made all data underlying the findings in their manuscript fully available?

Reviewer #1: Yes

Reviewer #2: Yes

5. Is the manuscript presented in an intelligible fashion and written in standard English?

Reviewer #1: Yes

Reviewer #2: Yes

6. Review Comments to the Author

Reviewer #1: I thank the authors for addressing both my conceptual and methodological concerns about this manuscript. I especially appreciated the inclusion of revised/additional analyses regarding the Moderna vaccine news, and the assessment of demographic balance across conditions. I found the updated manuscript to be significantly improved, and I now recommend publication.

Reviewer #2: (No Response)

7. PLOS authors have the option to publish the peer review history of their article (what does this mean?). If published, this will include your full peer review and any attached files.

Reviewer #1: **Yes: **Matt Motta

Reviewer #2: No

---

## [Editor Report · Acceptance letter]

11 May 2021

PONE-D-21-00518R1 

Factors indicating intention to vaccinate with a COVID-19 vaccine among older U.S. Adults 

Dear Dr. Navar:

I'm pleased to inform you that your manuscript has been deemed suitable for publication in PLOS ONE. Congratulations! Your manuscript is now with our production department. 

Kind regards, 

on behalf of

Dr. Jean-François Daoust 

Academic Editor

PLOS ONE